# An Epilepsy-Associated CILK1 Variant Compromises KATNIP Regulation and Impairs Primary Cilia and Hedgehog Signaling

**DOI:** 10.3390/cells13151258

**Published:** 2024-07-26

**Authors:** Ana Limerick, Ellie A. McCabe, Jacob S. Turner, Kevin W. Kuang, David L. Brautigan, Yi Hao, Cheuk Ying Chu, Sean H. Fu, Sean Ahmadi, Wenhao Xu, Zheng Fu

**Affiliations:** 1Department of Pharmacology, University of Virginia, Charlottesville, VA 22908, USA; afl4j@virginia.edu (A.L.); eam8cd@virginia.edu (E.A.M.); jst7ee@virginia.edu (J.S.T.); kwk5ny@virginia.edu (K.W.K.); sve3pe@virginia.edu (C.Y.C.); gbn8rm@virginia.edu (S.H.F.); sean.ahmadi@gwmail.gwu.edu (S.A.); 2Department of Microbiology, Immunology and Cancer Biology, University of Virginia, Charlottesville, VA 22908, USA; db8g@virginia.edu (D.L.B.); wx8n@virginia.edu (W.X.); 3Department of Biochemistry and Molecular Genetics, University of Virginia, Charlottesville, VA 22908, USA; yh8a@virginia.edu

**Keywords:** CILK1, KATNIP, epilepsy, variant, primary cilia, Hedgehog signaling

## Abstract

Mutations in human *CILK1* (ciliogenesis associated kinase 1) are linked to ciliopathies and epilepsy. Homozygous point and nonsense mutations that extinguish kinase activity impair primary cilia function, whereas mutations outside the kinase domain are not well understood. Here, we produced a knock-in mouse equivalent to the human *CILK1* A615T variant identified in juvenile myoclonic epilepsy (JME). This residue is in the intrinsically disordered C-terminal region of CILK1 separate from the kinase domain. Mouse embryo fibroblasts (MEFs) with either heterozygous or homozygous A612T mutant alleles exhibited a higher ciliation rate, shorter individual cilia, and upregulation of ciliary Hedgehog signaling. Thus, a single A612T mutant allele was sufficient to impair primary cilia and ciliary signaling in MEFs. Gene expression profiles of wild-type versus mutant MEFs revealed profound changes in cilia-related molecular functions and biological processes. The CILK1 A615T mutant protein was not increased to the same level as the wild-type protein when co-expressed with scaffold protein KATNIP (katanin-interacting protein). Our data show that KATNIP regulation of a JME-associated single-residue variant of CILK1 is compromised, and this impairs the maintenance of primary cilia and Hedgehog signaling.

## 1. Introduction

Most eukaryotic cells use a microtubule-based apical membrane protrusion called the primary cilium to sense environmental cues and transduce extracellular signals [1]. The primary cilium functions as a signaling hub, compartmentalizing ion channels, receptor tyrosine kinases, G-protein coupled receptors, and second messengers (e.g., calcium and cyclic AMP) [2]. Primary cilia defects impair the sensory and signaling functions of a cell, leading to abnormalities in tissue development and homeostasis that underlie a large category of human disorders referred to as ciliopathies [3,4,5]. Patients with ciliopathies exhibit tissue and skeletal deformities, cognitive deficits, and behavioral anomalies, implicating an important role of primary cilia as a non-synaptic mechanism to regulate neuronal circuits and functions [6].

One of the key regulators that controls primary cilia formation and elongation is CILK1 (ciliogenesis associated kinase 1) [7]. CILK1 consists of an N-terminal catalytic domain (residues 1–284) and a C-terminal domain with an intrinsically disordered region (IDR, residues 285–632). The scaffold protein KATNIP binds to the IDR and stabilizes and activates CILK1 [8]. The homozygous mutations (e.g., R272Q) in the catalytic domain of CILK1 abolished kinase activity, impaired cilia length control and Hedgehog signaling, and caused ciliopathy phenotypes [9]. How CILK1 dysfunction leads to ciliopathies is still poorly understood. CILK1 and its homologue DYF in *C. elegans* interact with intraflagellar transport (IFT) protein complexes to regulate ciliary trafficking [10,11,12,13,14,15], which is critical for cilia formation and elongation. Kinesin-2 motor protein KIF3A was identified as a CILK1 substrate [10,16]. However, elimination of the CILK1 phosphorylation of KIF3A by a point mutation was not sufficient to reproduce the severe ciliopathy phenotypes caused by CILK1 deletion or loss-of-function mutations, suggesting that some other substrate(s) besides KIF3A mediates CILK1 signaling to affect IFT and primary cilia [17].

The heterozygous variants in the IDR (e.g., K305T and A615T) are strongly linked to juvenile myoclonic epilepsy (JME) [18]. Ectopically expressed CILK1 variants in the mouse neocortex impaired radial migration and the cell cycle of neural progenitor cells, possibly through a dominant-negative action [18]. However, it remains to be determined whether these JME-associated heterozygous variants can impact primary cilia and by what mechanisms.

## 2. Materials and Methods

### 2.1. Reagents and Antibodies

pCMV6-Myc-Flag-CILK1 (RC213609) and pCMV6-Myc-Flag-KATNIP (RC220412) were from Origene (Rockville, MD, USA). CILK1 A615T mutant sequence was amplified by PCR and cloned into pCMV6-Myc-Flag vector [19]. Smo (E-5) mouse monoclonal antibody (sc-166685) and Gli-1 (C-1) mouse monoclonal antibody (sc-515751) were from Santa Cruz Biotechnology (Dallas, TX, USA). Beta-tubulin (9F3) rabbit monoclonal antibody (2128) and ERK1/2 rabbit polyclonal antibody (9102) were from Cell Signaling Technology (Danvers, MA, USA). Arl13B rabbit polyclonal antibody (17711-1-AP) and γ-tubulin mouse monoclonal antibody (66320-1-Ig) were from Proteintech (Rosemont, IL, USA). Goat anti-rabbit IgG (Alexa Fluor 488) antibody (ab150081) and goat anti-mouse IgG (Alexa Fluor 594) antibody (ab150120) were from Abcam (Cambridge, MA, USA). SAG (Smoothened Agonist) HCl (S7779) was from Selleckchem Chemicals LLC (Houston, TX, USA).

### 2.2. MEF Cell Isolation and Culture

MEF cells were isolated from E15.5 embryos and maintained at 37 °C and 5% CO_2_ in Dulbecco’s modified Eagle’s medium (DMEM) supplemented with 4.5 g/L glucose, 10% fetal bovine serum, and penicillin–streptomycin using a standard protocol [20].

### 2.3. Immunoblotting

Cells were lysed in lysis buffer (50 mM Tris-HCl, pH 7.4, 150 mM NaCl, 1% NP-40, 2 mM EGTA, complete protease inhibitors (Roche), 10 mM sodium orthovanadate, 5 mM sodium fluoride, 10 mM sodium pyrophosphate, 10 mM β-glycerophosphate, and 1 µM microcystin LR). Cell lysate was cleared by microcentrifugation. Cell extracts were boiled for 5 min in an equal volume of 2X Laemmli sample buffer and resolved by SDS–PAGE. Samples were transferred to a nitrocellulose membrane and blocked for one hour in 5% dry milk before an overnight primary antibody incubation in TBS containing 0.1% Tween-20 and 5% bovine serum albumin (BSA) at 4 °C. This was followed by multiple rinses and one-hour incubation with horseradish peroxidase (HRP)-conjugated secondary antibody. Chemiluminescence signals were developed using Millipore Immobilon ECL reagents.

### 2.4. Immunofluorescence

MEF cells grown on gelatin-coated coverslips were fixed by 4% paraformaldehyde in PBS, rinsed in PBS, and then permeabilized by 0.2% Triton X-100 in PBS. After one hour in blocking buffer (3% goat serum, 0.2% Triton X-100 in PBS), cells were incubated with primary antibodies at 4 °C overnight followed by rinses in PBS and one-hour incubation with Alexa Fluor-conjugated secondary antibodies. After multiple rinses, slides were mounted in antifade reagent containing DAPI (4′,6-diamidino-2-phenylindole) for imaging via a confocal Laser Scanning Microscopy 700 from ZEISS (Chester, VA, USA) at the UVA Advanced Microscopy Facility.

### 2.5. Cilia Length Measurement

The Zen 2009 program was used with a confocal Laser Scanning Microscope 700 from ZEISS (Chester, VA, USA) to collect z stacks at 0.5 μm intervals to incorporate the full axoneme based on immunostaining of cilia marker Arl13b and basal body marker γ-tubulin. All cilia were then measured in ImageJ via a standardized method based on the Pythagorean Theorem in which cilia length was based on the equation L^2^ = z^2^ + c^2^, in which “c” is the longest flat length measured from the z slices, and “z” is the number of z slices in which the measured cilia were present multiplied by the z stack interval (0.5 μm).

### 2.6. RNA-Seq Analysis and Enrichment Analysis

RNA-seq analysis was conducted at the University of Virginia, Genome Analysis and Technology Core, RRID:SCR_018883. RNA-Seq reads were aligned using STAR [21], with mouse genome build mm10 as reference. Expression count matrices were calculated using RSEM [22]. Differential expression analysis of RNA-seq was performed in R using the EdgeR package [23] with a Benjamini–Hochberg FDR of 0.001. Three replicates were sequenced and analyzed for each condition of WT and A612T mutant. Ontology (BP and MF) and KEGG pathway enrichment analysis were performed by Enrichr [24]. Differentially expressed genes with abs(log2FoldChange) ≥ 2 and FDR < 0.001 were used in the functional analysis. Volcano plot and enrichment plots were made in R (4.2.3).

### 2.7. Statistical Analysis

Experimental data were analyzed by the two-tailed Student’s *t*-test to compare the means of two groups, and *p* values less than 0.05 were considered significant. One-way ANOVA with post hoc Tukey HSD test was used to assess the significance of differences between pairs of group means. For the Tukey test, the significance level at alpha values of 0.05 and 0.01 was used for evaluation.

## 3. Results

### 3.1. Cells Expressing the Mouse Equivalent of Human Variant CILK1 A615T

To evaluate the impact of an epilepsy-associated mutation, *CILK1* A615T, on primary cilia and ciliary Hedgehog signaling, we used CRISPR/Cas9 to genetically engineer mutations in exon 13 of mouse *Cilk1* (Figure 1A). These knock-in mutations converted Ala612 (the mouse equivalent of human Ala615) to Thr and also introduced an Hpy99I restriction enzyme recognition site near the A612T mutation, as confirmed by sequencing results (Figure 1B). Restriction digestion of PCR products by Hpy99I allowed us to distinguish between *Cilk1* wild-type (WT) and A612T mutant alleles, as shown in the genotyping results (Figure 1C). The A612T heterozygous mice are viable and fertile. Breeding of heterozygotes produced offspring of three genotypes at the Mendelian ratio. Both heterozygotes and homozygotes appeared to be normal (e.g., body weight and movements) compared to their wild-type littermates. We then isolated mouse embryonic fibroblasts (MEFs) from E15.5 *Cilk1* WT and mutant embryos with heterozygous and homozygous A612T substitutions.

### 3.2. Cilk1 A612T Mutant Allele Induces Higher Ciliation Frequency but Shorter Cilia

We immunostained MEFs with primary cilia marker Arl13B and basal body marker γ-tubulin (Figure 2A), calculated the percent of ciliated cells, and measured cilia length. Our results revealed that about 20% of WT MEFs had cilia under standard growth conditions (Figure 2B). By comparison, MEFs carrying a heterozygous A612T mutation showed a statistically significant increase in the fraction of cells with cilia (ciliation rate) to nearly double the rate of WT MEFs (Figure 2B). With both copies of the CILK1 gene mutated in homozygous MEFs, there was a further increase in the fraction of cells with cilia, more than double compared to WT MEFs (Figure 2B). These results indicated a dose-dependent effect of the amount of mutated CILK1 required to alter the formation of primary cilia. On the other hand, the average cilia length was 2.2 µm in WT MEFs compared to 1.8 µm in A612T mutant MEFs, a statistically significant decrease of about 18% (Figure 2C). In this case, MEFs bearing a heterozygous A612T mutation were not significantly different from those bearing a homozygous A612T mutation. This demonstrates haploinsufficiency of the WT allele, indicating that the length of the primary cilium requires both WT copies of the gene.

### 3.3. Cilk1 A612T Mutant Allele Upregulates Hedgehog Signaling

One test of cilia function is Hedgehog (Hh) signaling. The binding of an Hh ligand to its receptor Patched 1 (Ptch1) relieves Ptch1-mediated inhibition of Smoothened (Smo), causing Smo to translocate to the primary cilium for activation of downstream effectors such as Gli1 and Gli2. Smo translocation and activation can be induced by Hh agonists such as SAG. We treated wild-type MEF cells with either DMSO as a control or SAG (50 nM and 200 nM) to activate Hh signaling (Appendix A). We observed a significant accumulation of Smo in the primary cilium upon treatment with 200 nM SAG, based on Smo co-localization with the cilia marker Arl13B (Figure 3A and Appendix A). In comparison to WT cells, both heterozygous and homozygous A612T mutant cells showed statistically significant upregulation of Smo and Gli1 upon SAG stimulation (Figure 3B,C). Although the signals were weaker without added SAG, we did observe an increase in Hedgehog signaling in the variant vs. WT cells. There was no significant difference in the protein levels of Smo and Gli1 between the heterozygous and the homozygous mutant cells (Figure 3C). These results indicated that one *Cilk1* A612T mutant allele was sufficient to upregulate the Hedgehog pathway, in particular in response to SAG stimulation.

### 3.4. Cilk1 A612T Variant Produced Alterations in Cilia-Related Receptor Tyrosine Kinase (RTK) Pathways

To further examine the effects of the Cilk1 variant A612T on signaling pathways, we applied RNA-seq analysis to WT and mutant MEFs to compare gene expression. A volcano plot of differentially expressed genes (DEGs) showed that 1286 DEGs were identified, including 931 upregulated genes and 355 downregulated genes (Figure 4A, Appendix A). We performed Gene Ontology (GO) enrichment analysis of DEG (Fold Change ≥ 2 and FDR < 0.001). Molecular function (MF) analysis revealed enrichment of gene expression changes in transmembrane receptor kinase activity (Figure 4B), which is consistent with the role of primary cilia in the compartmentalization and regulation of receptor tyrosine kinase pathways. Biological process (BP) analysis identified that these changes are mainly involved in the regulation of extracellular matrix organization, angiogenesis, and cell migration (Figure 4C). Importantly, both MF and BP analyses suggested that these changes are related to transmembrane receptor tyrosine kinase pathways and protein phosphorylation.

### 3.5. The A615T Variant Compromises KATNIP Regulation of CILK1

Recently, we showed that the scaffold protein KATNIP interacts with CILK1, increasing activation of the kinase and elevating levels of CILK1 in cells [8]. Given that the A615T mutation is located in the intrinsically disordered region of CILK1 that mediates interaction with KATNIP, we examined whether the A615T mutation affected KATNIP regulation of CILK1. We transfected HEK293T cells with Flag-CILK1 (WT or A615T) and co-transfected an increasing amount of Flag-KATNIP, and then quantified the protein levels of CILK1 on Western blots (Figure 5). KATNIP co-expression enhanced WT CILK1 levels more than 20-fold in a dose-dependent manner. In comparison, at the same doses, KATNIP was less than half as effective at increasing the protein level of the A615T mutant protein, a statistically significant difference (Figure 5B). This result indicates that KATNIP regulation of the CILK1 A615T variant protein was relatively deficient.

## 4. Discussion

We examined the effects of a knock-in single-residue substitution, A612T, in the non-catalytic region of CILK1 kinase on its function in MEFs. Previous studies of CILK1 null mice revealed that when one copy of the *Cilk1* gene was deleted, the remaining copy was sufficient to support a wild-type phenotype in MEFs. Thus, CILK1 shows haplosufficiency, and this would argue that reduced levels of the CILK1 protein are able to fulfill functions in MEFs. Accordingly, ciliopathy phenotypes in MEFs require deletion or inactivation of both *Cilk1* alleles [9,10,25]. By contrast, here we observed that, compared to WT MEFs, heterozygous *Cilk1A612T*^+/−^ MEFs exhibited shorter cilia and increased ciliation, which are phenotypes generated from the modification of only some of the CILK1 in cells that also express the WT protein. This is a curious observation that we do not yet fully understand. Somehow, the mutant CILK1 protein interferes with the function of the WT protein in the same cells to produce a cilia phenotype. One way of producing such a dominant-negative effect is to compete for a limiting regulatory protein partner. In the current case, this might be the activating subunit KATNIP.

Our data showed that when co-expressed with KATNIP, the CILK1 A615T mutant protein levels were significantly lower compared to the wild-type protein. This suggests that the A615T mutation prevented KATNIP stabilization of CILK1 (Figure 5). This could be due to weaker or non-productive binding of KATNIP to CILK1 A615T. Weaker interactions would not support dominant-negative effects; however, if the affinity of KATNIP binding to CILK1 A615T was the same or even higher than to the WT protein, then the mutant protein would deplete the KATNIP available for activating WT CILK1. These alternative mechanisms merit further investigation of the direct interaction between CILK1 and KATNIP using purified proteins that are not yet available. Of course, it is also possible that the CILK1 A615T mutant protein may gain functions yet to be identified that interfere with the control of cilia formation.

In a previous study, we overexpressed JME-associated CILK1 variants, including A615T, in NIH-3T3 cells [19]. Compared with untransfected cells, cells transfected with wild-type CILK1 displayed a marked reduction in both cilia frequency and length, consistent with CILK1 function as a negative regulator of primary cilia [19]. By contrast, the A615T variant caused a marked increase in ciliation rate and only a slight increase in cilia length, implicating that this variant can impose a dominant-negative effect on ciliogenesis in the presence of WT CILK1 [19]. Here, in our heterozygous knock-in cells, this endogenous variant also increased ciliation but decreased cilia length. There are at least two potential mechanisms that may explain this discrepancy in cilia length between the knock-in and the overexpression models. First, mislocalization due to overexpression may compromise the ability of the variant to compete with endogenous proteins for the regulation of cilia. Second, mislocalization may also enable the variant to gain functions that interfere with the control of cilia length. Using GFP fluorescence, we previously showed that CILK1 wild-type proteins were highly enriched at the cilia base, whereas A615T mutant proteins were localized throughout the cilia axoneme [19]. Currently, we do not have an antibody that is reliable for the detection of endogenous CILK1 localization by immunofluorescence; therefore, it remains to be determined if the endogenous A615T proteins are mislocalized in the primary cilium.

An analysis of differentially expressed genes (Figure 4) revealed the top three biological processes impacted by the mouse A612T mutation: extracellular matrix organization, regulation of angiogenesis, and regulation of cell migration. Disruption of the extracellular matrix (ECM) by ciliopathy mutations has been implicated in the pathogenesis of these diseases because the ECM provides structural support for the maintenance of tissue and cell homeostasis [26]. It is of immense interest to understand the impact of an epilepsy-associated variant on the ECM composition, remodeling, and reorganization. Vascular endothelial primary cilia function as molecular switches for calcium and nitric acid signaling. Dysfunction of endothelial cilia impairs blood flow sensing and thus contributes to cardiovascular diseases [27]. Primary cilia are pivotal to interneuron migration since disrupting ciliary functions in cortical interneurons interfered with the migratory process [28]. A recent study showed that rhythmic cell migration is dependent on the circadian oscillation of primary cilium length [29]. Our data supported the observation that the GFP-tagged human A615T variant interfered with the radial migration of neural progenitor cells [18]. In future studies, we will validate major hits in cilia-dependent transmembrane receptor kinase pathways and explore potential mechanisms connecting this disease mutation to the alterations in cilia-related biological processes.

Our data clearly demonstrated effects of the CILK1 A615T variant on primary cilia and cilia-dependent Hedgehog signaling in MEFs. What remains unknown is its possible impact on neuronal functions and seizures, which would be relevant to human JME where this variant was found. The A612T mice are viable and fertile, so we will have a cohort of animals to study. It is worth pointing out that alterations in primary cilia have been associated with a variety of neurological disorders [6,30,31]. For instance, diminished primary cilia formation has been observed in psychiatric diseases such as schizophrenia and bipolar disorders [32]. Both spontaneous and induced seizures have been shown to disrupt primary cilia growth and length control [33]. The primary cilium has been proposed as a possible marker for the diagnosis of neurological diseases. Perhaps pharmacological interventions that impact CILK1 will have some utility in therapeutics.

## Figures and Tables

**Figure 1 cells-13-01258-f001:**
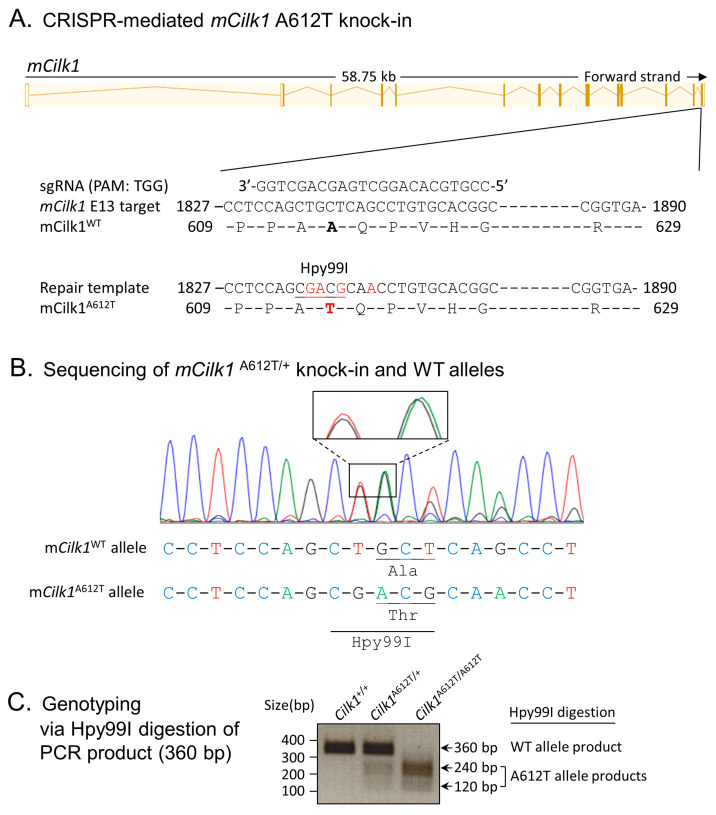
Generation of the *Cilk1* A612T knock-in mouse model. (**A**) A schematic illustration of CRISPR/Cas9-mediated knock-in of the A612T mutation and an Hpy99I restriction enzyme cutting site within the *mCilk1* mutant allele. (**B**) Sequencing results confirming designed mutations that not only converted Ala612 to Thr but also introduced a new Hpy99I site near A612T for genotyping. One silent base substitution (CAG→CAA) was also engineered to deter the recut of the repair template by Cas9 nuclease. (**C**) Genotyping results from Hpy99I-digested PCR products that can distinguish between the WT allele and the A612T mutant allele.

**Figure 2 cells-13-01258-f002:**
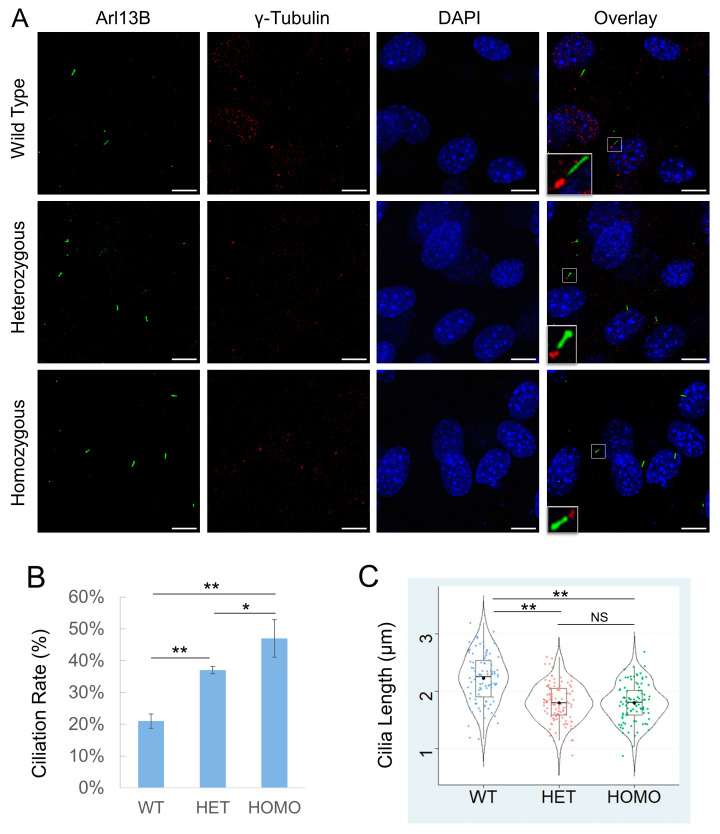
Effects of Cilk1 A612T mutant on primary cilia frequency and length. (**A**) Cilk1 WT and A612T mutant MEFs (mouse embryonic fibroblasts) were immunolabelled with the Arl13B antibody for primary cilia and the γ-tubulin antibody for basal bodies and stained with DAPI for nuclei. Scale bar, 5 µm. (**B**) Shown are ciliation rates of wild-type (WT, 1167 cells), A612T heterozygous (HET, 1295 cells), and homozygous (HOMO, 1428 cells) MEFs, mean ± SD, *n* = 3 independent experiments. (**C**) A violin box plot showing the distribution of numerical values of cilia length in Cilk1 WT control (*n* = 93 cilia), A612T HET (*n* = 109 cilia), and A612T HOMO (*n* = 94 cilia) MEFs, median with interquartile range box and min/max whiskers. One-way ANOVA followed by post hoc Tukey HSD test was used to assess the significance of differences between pairs of group means in both (**B**,**C**). The significance level was at alpha values of 0.05 (*) and 0.01 (**). NS = not significant.

**Figure 3 cells-13-01258-f003:**
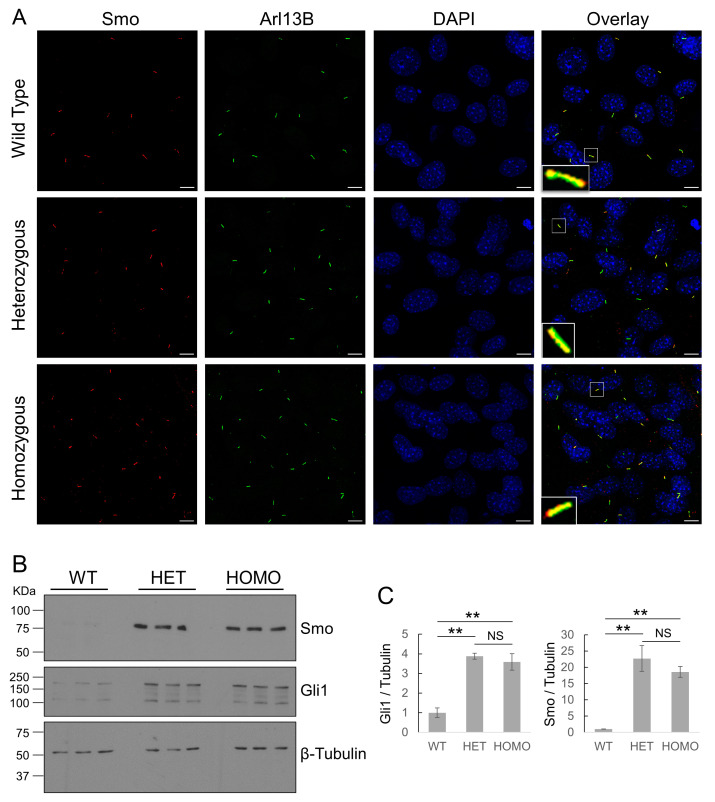
Effects of Cilk1 A612T on Hedgehog signaling. (**A**) MEFs were treated with Smo agonist SAG (200 nM) to activate Hedgehog signaling. Shown are MEFs immunolabelled with Smo and Arl13B antibodies and the overlay of Smo and Arl13B signals in primary cilia. DAPI labels the nucleus. Scale bar, 5 µm. (**B**) Equal amounts of total proteins extracted from MEFs treated with 200 nM SAG were blotted with Smo, Gli1, and β-tubulin (the loading control) antibodies. WT = Cilk1 wild type; HET = Cilk1 A612T heterozygous; HOMO = Cilk1 A612T homozygous. (**C**) Quantification of Gli1 and Smo signals on the Western blots in (**B**) relative to the loading control β-tubulin. Shown are fold changes relative to WT, means ± SD. One-way ANOVA followed by post hoc Tukey HSD test was used to assess the significance of differences between pairs of group means. The significance level was at alpha values of 0.01 (**). NS = not significant.

**Figure 4 cells-13-01258-f004:**
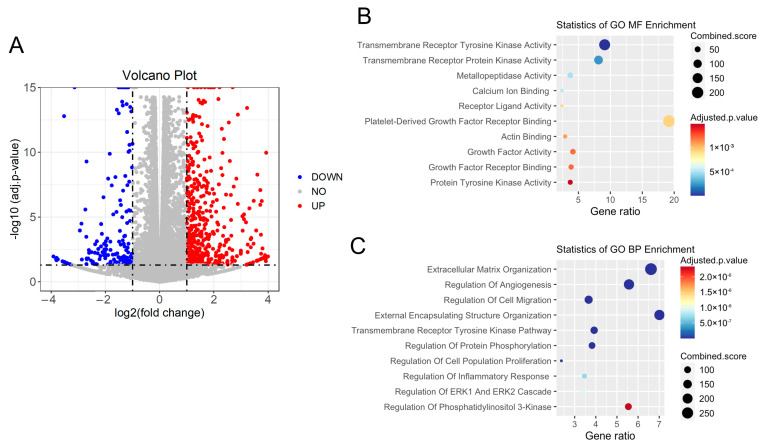
RNA-seq analysis of Cilk1 WT and A612T homozygous mutant MEFs. Total RNA was extracted and purified from MEF cells expressing wild-type (WT) or homozygous A612T mutant Cilk1. (**A**) The volcano plot showing differentially expressed genes (fold change > 2 or <0.5 and *p*-value < 0.05) in A612T mutant cells compared to WT cells. (**B**) The molecular function (MF) analysis of differentially expressed genes. (**C**) The biological process (BP) analysis of differentially expressed genes.

**Figure 5 cells-13-01258-f005:**
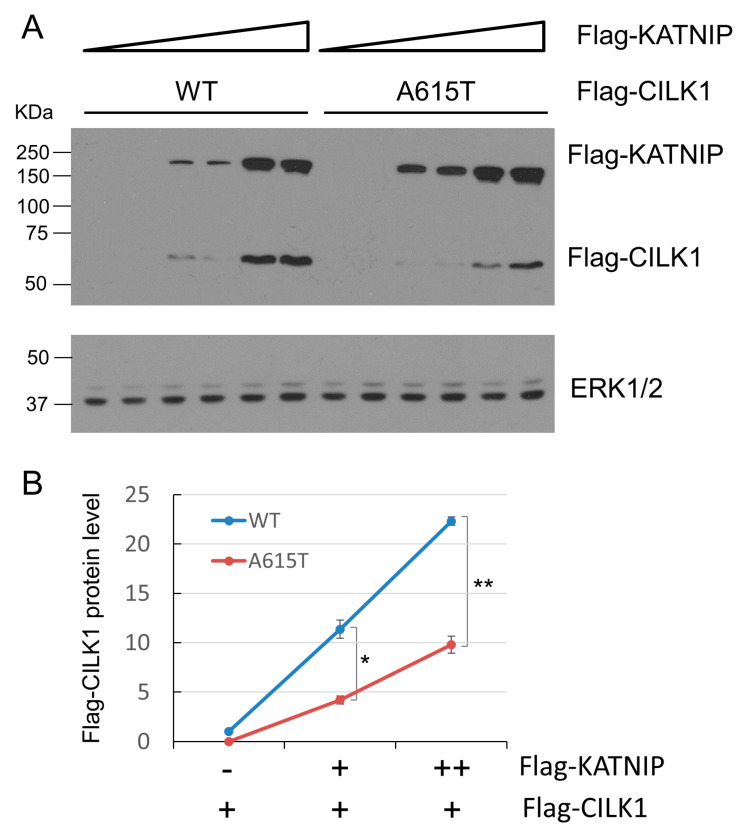
Effect of CILK1 A615T mutation on CILK1 stabilization by KATNIP. (**A**) Flag-CILK1 WT or A615T mutant was co-expressed with an increasing amount of Flag-KATNIP in HEK293T cells. Equal amounts of total proteins from cell lysates were blotted with Flag and ERK1/2 antibodies. (**B**) Flag-CILK1 proteins normalized against total ERK proteins were plotted as a function of the increasing level of Flag-KATNIP proteins. Quantification data were shown as mean ± SD, two-tailed Student *t*-test, * *p* < 0.05, ** *p* < 0.01. Shown here are the representative data from three independent experiments.

## Data Availability

The original contributions presented in the study are included in the article/Appendix A; further inquiries can be directed to the corresponding author.

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
