# Peer review of "An Epilepsy-Associated CILK1 Variant Compromises KATNIP Regulation and Impairs Primary Cilia and Hedgehog Signaling"

_cells, 2024, doi:10.3390/cells13151258_

Round 1

Reviewer 1 Report

Comments and Suggestions for Authors

Limerick et al., investigated the impact of introducing a single residue substitution A612T in the non-catalytic region of CILK1 kinase. They found the mutation impaired primary cilia and ciliary signaling in the embryo fibroblasts. Several concerns are listed below.

1. They generated a novel knock-in mice line but no characterization was reported. At least the weight, gender rate, genotype rate, basic behavior should be evaluated in CILK1 A612T mice.

2. Whether A615T mutation affects KATNIP interaction with CILK1 ? Co-ip experiment is needed.

3. They used flag antibody to recognize both Flag-KATNIP and Flag-CILK1. The blots should be in one image. The full length blot images are needed.

4. Although they showed A615T mutation induced shorter cilia and related protein and signaling impairment, there is lack of mechanism study on these changes.

Comments on the Quality of English Language

Minor editing of English language required

Author Response

We thank the reviewer for constructive comments that guided us in making revisions and additions that enhance the clarity and quality of this manuscript. Attached below please find a point-to-point response to the reviewer’s comments.

Reviewer #1

Limerick et al., investigated the impact of introducing a single residue substitution A612T in the non-catalytic region of CILK1 kinase. They found the mutation impaired primary cilia and ciliary signaling in the embryo fibroblasts. Several concerns are listed below.

1. They generated a novel knock-in mice line but no characterization was reported. At least the weight, gender rate, genotype rate, basic behavior should be evaluated in CILK1 A612T mice.

We agree with the reviewer, this information should have been included in the original manuscript and we have added a statement in the result section (Page 3, Lines 136-139).

2. Whether A615T mutation affects KATNIP interaction with CILK1? Co-IP experiment is needed.

We agree with the reviewer that this would be an informative result. However, we are unable to do this because of the lack of antibodies to effectively detect the endogenous proteins. We did the experiments in Fig. 5 as an approach to compare CILK1-KATNIP interactions.

Ideally, we need to measure the direct binding interaction between KATNIP and CILK1 WT and A612T. To do this, we need to separately purify KATNIP and CILK1 proteins. However, technically it is difficult to purify a sufficient amount of CILK1 WT or A615T mutant proteins and we will need some new approaches and extra time to accomplish this. We acknowledge this issue in the discussion (Page 9, Lines 237-239).       

3. They used flag antibody to recognize both Flag-KATNIP and Flag-CILK1. The blots should be in one image. The full length blot images are needed.

We have included a new full-length blot image in revised Fig. 5A (Page 8). 

4. Although they showed A615T mutation induced shorter cilia and related protein and signaling impairment, there is lack of mechanism study on these changes.

Recently, we have discovered that KATNIP functions as a scaffold of CILK1 to stabilize and activate CILK1 and enhance CILK1 control of cilia length. In the current manuscript, we showed clear evidence in Fig. 5 that the A615T mutation compromised KATNIP stabilization effect on CILK1 in cells, providing a potential mechanistic link to the cilia phenotype of the A615T variant. We agree with the reviewer that detailed mechanisms are yet to be provided to fully explain the molecular basis of the A615T phenotype, which requires extensive experimentation and is outside the scope of the current manuscript.    

Reviewer 2 Report

Comments and Suggestions for Authors

The manuscript by Limerick et al focuses on the effect of a variant in ciliogenesis associated kinase 1 (CILK1) in ciliogenesis. This variant (A615T) is in the intrinsically disordered region (IDR) of CILK1, which can bind to katanin-interacting protein (KATNIP) to control its stability. This CILK1 variant is linked to juvenile myoclonic epilepsy (JME), but the mechanistic link between this variant and JME is unknown. The authors show that this CILK1 A615T variant leads to increased ciliation, decreased cilia length, and enhanced Hedgehog signaling. They also propose that this abnormal phenotype is due to the inability of the scaffolding protein, KATNIP, to bind to this CILK1 variant. However, the manuscript lacks some experimental details and discussion points, as described below.     

Major Issues:

1.       Figure 2: Is Panel A an image from a WT or mutant MEFs? Please include representative images that show increased ciliation and decreased cilia length that correspond to WT, HET, and HOMO MEFs.

2.       Figure 2: In Panel B, since more than two groups are being analyzed, a one-way ANOVA followed by a post-hoc test should be used as was done in Panel C.

3.       Figure 3: Are Panels A-C from WT or mutant MEFs? Please include representative images that show increased ciliary Smo in HET and HOMO MEFs.

4.       Figure 3: It was concluded that HET and HOMO MEFs have increased Hh signaling based on the Western blot. However, the loading control (β-Tubulin) for HOMO is higher than the WT and HET conditions. Please include quantification for the relative levels of Smo and Gli1 in the HET and HOMO conditions compared to WT. It was also stated that “there was no major difference in the protein level of Smo and Gli between the heterozygous and homozygous mutant cells.” This quantification would show this conclusion.

5.       Figure 3: It was stated that “Cilk1 A612T mutant allele was sufficient to up-regulate the Hedgehog pathway.” The data is only shown for Cilk1 variants in the presence of SAG. Was this Cilk1 mutant allele sufficient for upregulating Hedgehog activity without SAG?

6.       Figure 4: Although this data is interesting, it seems out of place. The results from this experiment are never mentioned again. Several genes that are altered in major categories (ie. transmembrane receptor tyrosine and protein kinase activity) should be validated and their potential mechanism should be discussed in the Discussion section. Also, were the MEFs used in this experiment HET or HOMO MEFs?  

7.       Figure 5: Although increasing KATNIP levels don’t increase CILK1 A615T protein levels like CILK1 WT, there is no direct evidence showing that KATNIP and CLIK1 directly interact. Is there decreased binding of KATNIP to CILK1 A615T relative to WT CILK1 to provide evidence of KATNIP acting as a scaffold to control the stability of CILK1?

Minor Issues:

1.       Introduction: Please expand on the known mechanisms of how CILK1 can lead to ciliopathies and JME.

2.       Methods: In the “Reagents and Antibodies” subsection (2.1), PDGFRβ is listed as an antibody, but it was not used in the study.

3.       Figure 1: Please provide a representative blot of Cilk1 A612Y homozygous mutant genotyping results

4.       Discussion: Previous studies by the lab (Wang EJ et al. Cells. 2020 – reference 11) have shown that this A615T variant increases cilia length and ciliation, but here it is shown that this variant increases ciliation but decreases length. Please discuss potential mechanisms for this discrepancy in cilia length and how results from this knock-in model would differ from the overexpression model. If there is a dominant negative phenotype as suggested, then overexpression of this variant should exhibit a similar phenotype as the heterozygous knock-in model.

Author Response

We thank the reviewer for constructive comments that guided us in making revisions and additions that enhance the clarity and quality of this manuscript. Attached below please find a point-to-point response to the reviewer’s comments.

Reviewer #2

The manuscript by Limerick et al focuses on the effect of a variant in ciliogenesis associated kinase 1 (CILK1) in ciliogenesis. This variant (A615T) is in the intrinsically disordered region (IDR) of CILK1, which can bind to katanin-interacting protein (KATNIP) to control its stability. This CILK1 variant is linked to juvenile myoclonic epilepsy (JME), but the mechanistic link between this variant and JME is unknown. The authors show that this CILK1 A615T variant leads to increased ciliation, decreased cilia length, and enhanced Hedgehog signaling. They also propose that this abnormal phenotype is due to the inability of the scaffolding protein, KATNIP, to bind to this CILK1 variant. However, the manuscript lacks some experimental details and discussion points, as described below.     

Major Issues:

1. Figure 2: Is Panel A an image from a WT or mutant MEFs? Please include representative images that show increased ciliation and decreased cilia length that correspond to WT, HET, and HOMO MEFs.

We have added more representative images, as requested, in the new Fig. 2A (Page 5).

2. Figure 2: In Panel B, since more than two groups are being analyzed, a one-way ANOVA followed by a post-hoc test should be used as was done in Panel C.

Correct. Good suggestion. We have applied, as directed, one-way ANOVA followed by a post-hoc test in Fig. 2B (Page 5).

3. Figure 3: Are Panels A-C from WT or mutant MEFs? Please include representative images that show increased ciliary Smo in HET and HOMO MEFs.

We have added new representative images from both WT and mutant MEFs in the revised Fig. 3A (Page 6). There is a large variation in the intensity of Smo signals in WT and mutant MEFs, so it is not feasible to compare Smo signal intensity in individual cilia between WT and mutant MEFs. Therefore, Western blots with quantification data in Fig. 3B-C are used to evaluate the overall changes in Smo levels, which should reflect the overall difference in the sum of the total number of cilia and the signal level in each cilium.    

4. Figure 3: It was concluded that HET and HOMO MEFs have increased Hh signaling based on the Western blot. However, the loading control (β-Tubulin) for HOMO is higher than the WT and HET conditions. Please include quantification for the relative levels of Smo and Gli1 in the HET and HOMO conditions compared to WT. It was also stated that “there was no major difference in the protein level of Smo and Gli between the heterozygous and homozygous mutant cells.” This quantification would show this conclusion.

We agreed with the reviewer’s concerns and added quantification data in the new Fig. 3C (Page 6). Our new data support the conclusion that there was no significant difference in the protein level of Smo and Gli between the heterozygous and homozygous mutant cells.

5. Figure 3: It was stated that “Cilk1A612T mutant allele was sufficient to up-regulate the Hedgehog pathway.” The data is only shown for Cilk1 variants in the presence of SAG. Was this Cilk1 mutant allele sufficient for upregulating Hedgehog activity without SAG?

Yes, this mutant allele was sufficient for upregulating Hedgehog activity without SAG, but the signals were weaker than with SAG. We added a statement on page 7 (lines 176-178, 181) to include this observation and qualified the conclusion in results as “in particular in response to SAG stimulation.”

6. Figure 4: Although this data is interesting, it seems out of place. The results from this experiment are never mentioned again. Several genes that are altered in major categories (i.e. transmembrane receptor tyrosine and protein kinase activity) should be validated and their potential mechanism should be discussed in the Discussion section. Also, were the MEFs used in this experiment HET or HOMO MEFs?

We thank the reviewer for this point. We have included a new paragraph in the discussion (Page 9, Lines 262-278) to address the significance of the data and future directions.   

7. Figure 5: Although increasing KATNIP levels don’t increase CILK1 A615T protein levels like CILK1 WT, there is no direct evidence showing that KATNIP and CLIK1 directly interact. Is there decreased binding of KATNIP to CILK1 A615T relative to WT CILK1 to provide evidence of KATNIP acting as a scaffold to control the stability of CILK1?

We agree with the reviewer that measuring direct interaction between KATNIP and CILK1 is necessary to show decreased binding of KATNIP to CILK1 A615T. To do this, we need to separately purify KATNIP and CILK1 proteins. Technically it is difficult to purify a sufficient amount of WT and A615T mutant proteins. Currently we are searching for a biochemical strategy to overcome this hurdle. We acknowledged this point of view in the discussion (Page 9, Lines 237-239).     

Minor Issues:

1. Introduction: Please expand on the known mechanisms of how CILK1 can lead to ciliopathies and JME.

Thank you for this suggestion. Please see new information (highlighted in red) in the introduction on page 2 and lines 48-56.  

2. Methods: In the “Reagents and Antibodies” subsection (2.1), PDGFRβ is listed as an antibody, but it was not used in the study.

Sorry for this error, it is now deleted.

3. Figure 1: Please provide a representative blot of Cilk1A612T homozygous mutant genotyping results

Please see the revised Fig. 1C (Page 4) for this new addition. 

4. Discussion: Previous studies by the lab (Wang EJ et al. Cells. 2020 – reference 11) have shown that this A615T variant increases cilia length and ciliation, but here it is shown that this variant increases ciliation but decreases length. Please discuss potential mechanisms for this discrepancy in cilia length and how results from this knock-in model would differ from the overexpression model. If there is a dominant negative phenotype as suggested, then overexpression of this variant should exhibit a similar phenotype as the heterozygous knock-in model.

Thank you for this suggestion. Please see a new paragraph (highlighted in red) in the discussion on page 9 and lines 243-260.

Round 2

Reviewer 1 Report

Comments and Suggestions for Authors

N/A

Reviewer 2 Report

Comments and Suggestions for Authors

The manuscript by Limerick et al focuses on the effect of a variant in ciliogenesis associated kinase 1 (CILK1) in ciliogenesis. This variant (A615T) is in the intrinsically disordered region (IDR) of CILK1, which can bind to katanin-interacting protein (KATNIP) to control its stability. This CILK1 variant is linked to juvenile myoclonic epilepsy (JME), but the mechanistic link between this variant and JME is unknown. The authors show that this CILK1 A615T variant leads to increased ciliation, decreased cilia length, and enhanced Hedgehog signaling. They also propose that this abnormal phenotype is due to the inability of the scaffolding protein, KATNIP, to bind to this CILK1 variant.

The authors have addressed the major and minor concerns by adding further details about data visualization and analysis as well as the discussion of conclusions.